# Water Stress Promotes Secondary Sexual Dimorphism in Ecophysiological Traits of Papaya Seedlings

**DOI:** 10.3390/plants14152445

**Published:** 2025-08-07

**Authors:** Ingrid Trancoso, Guilherme A. R. de Souza, João Vitor Paravidini de Souza, Rosana Maria dos Santos Nani de Miranda, Diesily de Andrade Neves, Miroslava Rakocevic, Eliemar Campostrini

**Affiliations:** 1Setor de Fisiologia Vegetal, Laboratório de Melhoramento Genético Vegetal, Centro de Ciências e Tecnologias Agropecuárias, Universidade Estadual do Norte Fluminense (UENF), Avenida Alberto Lamego 2000, Parque Califórnia, Campos dos Goytacazes 28013-602, RJ, Brazil; ingrid-trancoso@hotmail.com (I.T.); guilherme.rodrigues@edu.uniube.br (G.A.R.d.S.); joaovitor.paravidini@unicatt.it (J.V.P.d.S.); nani@pq.uenf.br (R.M.d.S.N.d.M.); diesilyandrade@gmail.com (D.d.A.N.); campostenator@gmail.com (E.C.); 2Department of Sustainable Crop Production, Università Cattolica del Sacro Cuore, 29122 Piacenza, Italy

**Keywords:** biomass, *Carica papaya* L., gender segregation, morphology, optical leaf properties, SSD, SPAD, stomatal conductance

## Abstract

Plant genders could express different functional strategies to compensate for different reproductive costs, as females have an additional role in fruit and seed production. Secondary sexual dimorphism (SSD) expression is frequently greater under stress than under optimal growth conditions. The early gender identification in papaya may help to reduce orchard costs because the most desirable fruit shape is formed by hermaphrodite plants. We hypothesized that (a) gender ecophysiological phenotyping can be an alternative to make gender segregations in papaya seedlings, and (b) such gender segregation will be more efficient after a short drought exposure than under adequate water conditions. To test such hypotheses, seedlings of two papaya varieties (‘Candy’ and ‘THB’) were exposed to two kind of treatments: (1) water shortage (WS) for 45 h, after which they were well watered, and (2) continuously well-watered (WW). Study assessed the ecophysiological responses, such as stomatal conductance (*g*_s_), SPAD index, optical reflectance indices, morphological traits, and biomass accumulation in females (F) and hermaphrodites (H). In WS treatment, the SSD was expressed in 14 of 18 traits investigated, while in WW treatment, the SSD was expressed only in 7 of 18 traits. As tools for SSD expression, *g*_s_ and simple ratio pigment index (SRPI) must be measured on the first or second day after the imposed WS was interrupted, respectively, while the other parameters must be measured after a period of four days. In some traits, the SSD was expressed in only one variety, or the response of H and F plants were of opposite values for two varieties. The choice of the clearest responses of gender segregation in WS treatment will be greenness index, combination of normalized difference vegetation index (CNDVI), photochemical reflectance index (PRI), water band index (WBI), SRPI, leaf number, leaf dry mass, and leaf mass ratio. If the WW conditions are maintained for papaya seedling production, the recommendation in gender segregation will be the analysis of CNDVI, carotenoid reflectance index 2 (CRI2), WBI, and SRPI. The non-destructive optical leaf indices segregated papaya hermaphrodites from females under both water conditions and eventually could be adjusted for wide-scale platform evaluations, with planned space arrangements of seedlings, and sensor’s set.

## 1. Introduction

Only about 6% of higher plant species have separate genders in different individuals, a phenomenon known as dioecy [1]. In dioecious plant species, gender determination is usually possible only in the reproductive stage, while various plant characters are used for early determination in the seedling stage [2,3,4]. This determination is usually easier with the presence of X and Y chromosomes, but it is very rare among plant species [5].

Due to additional roles in fruit and seed production, female plants often invest more resources in reproduction than males, which generally results in different resource allocation trade-offs between plant functions for plant genders [6]. Plant genders could express different functional strategies to compensate for different reproductive costs. These strategies can be morphological/architectural [4,7], physiological [8,9,10], or metabolic [11], expressed as secondary sexual dimorphism (SSD). SSD represents a segregation among individuals of different genders in characteristics other than reproductive organs [12].

Genders usually express morphological and functional differences even before reproductive maturity [4], indicating that SSD expression is not only a consequence of resource allocation trade-offs between plant functions but also can have an evolutionary basis [13]. In some species, genders may differ functionally even under optimum environmental conditions [9], while gender differences are frequently greater under stressful conditions, but even under such conditions, the probability of SSD expression stays low [14]. It is observed that dioecy is associated with high genetic diversity and adaptation rates [15], which can explain a greater gender segregation under environmental pressures [14]. This means that SSD expressions of different physiological and morphological characters can be modified due to environmental conditions [9,16,17] and can vary over seasons and phenophases [18,19,20,21].

Let us see what happens with SSD in various dioecious species under water stress. Under such conditions, higher stomatal conductance (*g*_s_) and reduced leaf water potential are more commonly observed in females compared to males of *Salix glauca* [18]. Under a soil water deficit, the SSD is expressed in a lower net CO_2_ assimilation rate (*A*_net_) and *g*_s_ in females than in males of *Pistacia lentiscus* [22]. Under water stress, higher growth, chlorophyll (Chl) concentration, *A*_net_, *g*_s_, water use efficiency, photochemical, and non-photochemical quenching are more observed in females than in males of *Ginkgo biloba* [23]. Under the low water availability during the spring and autumn growth flushes, female plants of *Ilex paraguariensis* are more sensitive than male ones in leaf area and metamer number formation [16], but females show a higher Chl *b* index, photosynthetic capacity (*A*_max_) [20], and *A*_net_ [21] than males in early vegetative phases after pruning. Such female efforts could be considered as a general strategy to fulfill their additional effort in producing fruits and seeds in the latter phases [6].

In this paper, we will focus on SSD expressed in *g*_s_, relative Chl content, optical leaf properties, and morphology. The optical leaf properties and photosynthetic performance of a given leaf are determined by the structural leaf elements, with some of these having evolved as adaptations to environmental variations [24]. Due to variations in abiotic conditions, the genders of *Pistacia lentiscus* [19], *Amaranthus palmeri* [25], *Juniperus communis* [9], and *Ilex paraguariensis* [20] express SSD in their photochemical performance and photosynthetic pigment concentrations. In some morphological stages, optical leaf properties differ between the genders of *Ginkgo biloba* [26]. For example, the leaf reflectance of *Ginkgo biloba* is much higher in males than in females in the yellow leaf stage when the whole wavelength range spectrum is considered, while the reflectance of green leaves is very similar between the genders for the ranges of 437 to ~500 nm and 520 to ~640 nm.

Papaya (*Carica papaya* L.) is a polygamous (trioecious) species, forming male, female, and hermaphrodite flowers on different plants [3]. It is also agronomically interesting, as it is an important tropical fruit tree. The hermaphrodite plants are commercially the most desired, producing small-to-large fruits of good quality, due to their high pulp-to-volume ratio and the small cavities where the seeds are located [27,28]. Female plants produce medium-to-big, round-shaped fruits of good quality, but with a large cavity. Curiously, on the male plants, it is even possible to sporadically find some hermaphrodite flowers, which can produce elongated, pear-shaped fruits, but with low organoleptic quality [27].

In a commercial papaya plantation establishment, 2–4 seedlings are deposited per hole to obtain one hermaphrodite plant [2,29,30,31]. Then, 3–4 months later, when the plant genders can be identified based on the flower’s characteristics, hermaphrodite plants are preserved, and female and male plants are removed from the hole. In such management systems, the young plants are grown under competition for resources (water, mineral nutrients, and light) that can cause increased plant height, reduced stem diameter, and delayed beginning of the reproductive phase [29,31]. Papaya is sensitive to soil water deficits in the early-seedling stage [32] and even in adult plants, with expressive variety differences [33].

When papaya seedlings, previously defined as hermaphrodite, are taken to the field or indoor cultivations and planted with only one plant per hole, they show faster growth and higher stem diameter compared to the plants grown in a traditional competitive system of 2–4 seedlings per hole [30]. Thus, the early papaya gender identification may be an efficient way for commercial yield increases, reducing production costs, saving in labor, demanding a smaller number of seeds, and lower demands for inputs, such as water and mineral nutrients [34].

Papaya possesses homomorphic X and Y sex chromosomes, with a short non-recombining region on the Y chromosome, in the MSY region, indicating that the primitive papaya Y chromosome represents an early event in sex chromosome evolution [5]. In the mitotic metaphase plates, a homomorphic pair of sex chromosomes could not be distinguished in papaya, as is possible in some other species, such as *Rumex acetosa* [5]. Considering such knowledge, the gender chromosomes of papaya can be detected in the early juvenile phase using other genotyping techniques, such as molecular markers [35]. Although the technique of gender detection by molecular markers has been used for nearly 20 years in papaya, it increases production costs and is still not economically justified at the commercial level [34]. Additionally, some techniques, such as near-infrared spectroscopy associated with multivariate techniques, are shown to be successful as potential tools to detect genders in papaya seedlings and seeds [36]. Also, ecophysiological traits, such as Chl concentration, *A*_net_, and *g*_s_, are higher in leaves of adult female papaya plants than hermaphrodite papaya plants [37].

We hypothesized that: (a) gender ecophysiological phenotyping can be an alternative to make gender segregations in papaya seedlings, by using parameters such as *g*_s_, SPAD index, and leaf reflectance indices, and (b) such gender segregation will be more efficient when seedlings experience a short water shortage than under adequate water conditions. To test such hypotheses, seedlings of two papaya varieties (‘Candy’ and ‘THB’) were grown under well-watered (WW) and water-shortage (WS) treatments.

## 2. Results

### 2.1. SSD in Stomatal Conductance After a Water-Shortage Period

The *g*_s_ was gradually reduced from the first to the fourth day after the WS interruption, and such a dynamic was strongly dependent on the variety (Figure 1). ‘THB’ was the only variety reducing *g*_s_ after a water shortage (Figure 1), while ‘Candy’ was not sensitive to the imposed short-drought period. In the WW treatment, two varieties had similar *g*_s_. As *g*_s_ is one of the quickest ecophysiological responses regulating a chain of responses to drought, the SSD in *g*_s_ was observed on the first day after the water-shortage interruption in ‘THB’ (H > F) in WS treatment and even in WW in the ‘Candy’ genotype (F > H).

### 2.2. SSD in SPAD and Leaf Spectral Reflectance Indices After a Water-Shortage Period

#### 2.2.1. SSD in SPAD and Spectral Indices Associated with Chl Content

The SPAD was higher in ‘THB’ than in ‘Candy’ during four days of observation, irrespective of the water conditions (Figure 2A). For the ‘THB’, this variable remained similar between the WW and WS conditions, while the SPAD in ‘Candy’ was reduced in WS treatment from the first to fourth day after the water-shortage interruption. On the last evaluation day (45 DAS), the SSD in SPAD index was observed, but only in ‘Candy’ and only in WS treatment, being reduced by about 16% in the H plants compared to the F plants.

The greenness index (G index) responses were strongly impacted by variety during the days of evaluation (Figure 2B). ‘Candy’ showed higher values of the G index than ‘THB’ on the first and second day after the water-shortage interruption, irrespective of water treatment. On the second day after the water-shortage interruption (43 DAS), WS induced a higher G index than in WW treatment of ‘Candy’ variety, while the water regime did not modify this index in ‘THB’. A similar situation occurred on the third day after the water-shortage interruption (44 DAS), but only in females of ‘Candy’ that showed higher values of the G index in the WS treatment than in the WW treatment. On 44 DAS, F seedlings had a higher G index than H in ‘Candy’ in the WS treatment, while the opposite SSD expression was observed in ‘THB’. On the last observed day (45 DAS) and only in WS treatment, H seedlings of both varieties showed a higher G index than F seedlings, with ‘THB’ having a lower G index than ‘Candy’.

The combination of normalized difference vegetation index (CNDVI) was modified by three investigated factors (Figure 2C). Opposite to the G index (Figure 2B), it had lower values in the ‘Candy’ than in the ‘THB’ variety in WS treatment on the first day after the water-shortage interruption (Figure 2C). CNDVI values were higher in WS than in WW treatment in ‘THB’, while the opposite situation occurred in ‘Candy’ on the second day after the water-shortage interruption (43 DAS). Among the three Chl-associated indices investigated here (Figure 2), the CNDVI permitted the earliest SSD detection, on the second day after the water-shortage interruption, when females of both varieties had higher values than hermaphrodites in WS treatment, indicating that the latter gender was more sensitive to a short drought period than females (Figure 2C). A similar situation was also observed on 44 DAS (third day after the water-shortage interruption), when F seedlings showed higher CNDVI than H ones, even in WW treatment. On that DAS, the ‘THB’ had higher CNDVI than ‘Candy’, only in WW. Finally, on the fourth day after the water-shortage interruption (45 DAS), the SSD in CNDVI was observed in both WW and WS treatments, but with opposite responses: F seedlings had lower values than H in WW, while hermaphrodites were more sensitive to the short drought period than females. As expected, the CNDVI was higher in WW than in WS treatment.

#### 2.2.2. SSD in Spectral Indices Associated with Carotenoid and Flavonoid Contents

The carotenoid reflectance index 1 (CRI1), related to the total carotenoid (Car) and Chl contents, was dependent on water conditions on the first and second day after the WS interruption, when CRI1 was higher in WW than in WS treatment and opposite, respectively (Figure 3A). On the second and third day after the WS interruption (43 and 44 DAS), the ‘Candy’ variety showed higher CRI1 values than ‘THB’. On the fourth day after the WS interruption (45 DAS), a higher CRI1 was found in F than in H seedlings of ‘Candy’ grown in WW treatment, which was higher than F of the ‘THB’ variety. On that DAS, female ‘Candy’ seedlings showed superior CRI1 values than ‘Candy’ hermaphrodites in WS treatment, while the opposite SSD expression was observed in ‘THB’, where females showed lower CRI1 values than ‘THB’ hermaphrodites.

The carotenoid reflectance index 2 (CRI2), related to the total Car content, was dependent on water conditions and variety from 42 to 45 DAS (Figure 3B). On the first day after the WS interruption (42 DAS), CRI2 was higher in WW than WS treatment, and in ‘THB’ than in ‘Candy’ variety. On the second day after the WS interruption (43 DAS), the variety difference in CRI2 was only observed in WS treatment, where ‘Candy’ showed higher CRI2 values than ‘THB’. On that DAS, CRI2 values were higher in the WS than the WW treatment, probably due to the protective role of Car under drought. On the third day after the WS interruption (44 DAS), the hermaphrodites of both varieties had lower CRI2 in WW than WS treatment. On that DAS, the SSD in CRI2 was expressed only in ‘THB’ and only in WS treatment, with hermaphrodites showing higher values than females. On the fourth day after the WS interruption (45 DAS), the CRI2 values were reduced compared to those of 42 DAS. The SSD in CRI2 was expressed even under WW conditions on 45 DAS, where females had higher CRI2 than hermaphrodites. The opposite response occurred under the WS conditions, where SSD in CRI2 was only expressed in ‘THB’, with females having lower CRI2 than hermaphrodites.

The flavonoid reflectance index (FRI), related to flavonol pigments and their role in screening the excessive visible light and UV-A radiation, was impacted by water conditions (during all four days of measurements), and by variety (up to the third day after the water-shortage interruption) (Figure 3C). On the first day after the water-shortage interruption, ‘THB’ showed superior FRI values than ‘Candy’ in WW treatment, while such variety differentiation was not observed in WS treatment. On the second day after the water-shortage interruption, the FRI of WS seedlings was significantly higher than that of WW seedlings (about 10%), probably due to the developed flavonoid-protective role under the water stress. Such differentiation in FRI values was also observed on the third day after the water-shortage interruption, together with much higher FRI values in ‘Candy’ than in ‘THB’ in WS treatment. On the fourth day after the water-shortage interruption, the FRI was higher in WS than in WW by ~12%. The SSD was not expressed in the actual FRI analysis of papaya.

#### 2.2.3. SSD in Spectral Indices Associated with Water Status

The photochemical reflectance index (PRI), related to water status and an indicator of water stress, was dependent on all three factors studied here, starting from the first day after the water-shortage interruption, 42 DAS (Figure 4A). On that DAS, the greatest sensitivity to WS was observed in H seedlings of ‘Candy’, which showed lower PRI values than F seedlings. On the second day after the water-shortage interruption, ‘Candy’ WS seedlings showed lower PRI values than WW ones, and those values were much lower (~45%) than those of ‘THB’ in WS treatment. In WS treatment, on the third day, the PRI response of varieties was opposite to those on the second day after the water-shortage interruption: PRI values in ‘Candy’ were higher than in ‘THB’. On the third day, H seedlings of both varieties had lower PRI than F seedlings, and such SSD was expressed only under WS conditions. On the fourth day after the water-shortage interruption, the ‘THB’ variety showed higher PRI values than ‘Candy’ in WS (as on the second day), and under such conditions, the SSD was also expressed, with hermaphrodites being more sensitive to drought than females, as on the third day.

The water band index (WBI), related to water status and relative water content in leaves, was modified by water conditions and variety on the first day after the water-shortage interruption, 42 DAS (Figure 4B). On that DAS, the WBI values were lower in WS than WW treatment (as expected), and the ‘THB’ showed a lower WBI than ‘Candy’ under both water conditions. Surprisingly, on 43 and 44 DAS (second and third day after the water-shortage interruption), WBI had stable values, with no impact on any of the three factors studied. On the fourth day after the water-shortage interruption, the ‘Candy’ variety showed higher WBI values than ‘THB’ in WW, with hermaphrodites expressing lower WBI than females. On that day, the same SSD expression in WBI was observed under the WS conditions, but ’Candy’ was shown as a more sensitive variety to restrictive water conditions than ‘THB’.

#### 2.2.4. SSD in Spectral Indices Associated with Structure

The structure-insensitive pigment index (SIPI) maximizes the sensitivity to the bulk carotenoids-to-chlorophyll ratio and minimizes the impact of the canopy structure or leaf area index variables. On the first day after the water-shortage interruption (42 DAS), ‘Candy’ showed a lower SIPI than ‘THB’ only in the WS treatment (Figure 5A). On the second day after the water-shortage interruption (43 DAS), a similar comparison between the varieties was observed in WS treatment, while the opposite response of varieties was observed in WW treatment (‘Candy’ expressed higher SIPI than ‘THB’). On the third assessed day (44 DAS), female ‘Candy’ seedlings showed higher SIPI than hermaphrodite ones in WW, while the SSD expression was opposite in WS treatment (hermaphrodite ‘Candy’ seedlings had a higher SIPI than female ones). On the fourth day after the water-shortage interruption (45 DAS), ‘Candy’ continued to express the SSD in SIPI, which was higher in females than in hermaphrodites, under both water conditions.

The simple ratio pigment index (SRPI) is correlated to plant N status on large-area monitoring. On the first day after the water-shortage interruption (42 DAS), ‘Candy’ showed a higher SRPI than ‘THB’, irrespective of water conditions (Figure 5B). On that DAS, ‘Candy’ had a higher SRPI in WW than in WS treatment, while the situation was opposite for ‘THB’, which showed a higher SRPI in WS than in WW treatment. On the second day after the water-shortage interruption (43 DAS), only the significant gender effect was observed, with hermaphrodites having higher SRPI than females. No effect of any investigated factor was observed on the third day after the water-shortage interruption, while only the water regime effect was significant on the fourth day, with a higher SRPI in WW than in WS treatment.

### 2.3. SSD in Morphological Traits and Biomass Accumulation

The papaya seedling leaf number (LN) was reduced in WS when compared to WW conditions in both genders (F and H) of ‘THB’, but only in H of ‘Candy’ (Figure 6A). The SSD in LN was expressed in both varieties, but only in WS treatment, where hermaphrodites were more sensitive to drought than females. The papaya seedling stem diameter (SD) was reduced in WS compared to WW treatment, but no SSD was expressed in this morphological trait (Figure 6B).

Leaf dry mass (LDM) was higher in ‘THB’ than in ‘Candy’ under non-limited water conditions, while no variety differentiation was observed in WS treatment (Figure 6C). SSD was expressed in LDM, with females of both varieties producing higher LDMs than hermaphrodites, irrespective of water availability. Stem dry mass (SDM) was only impacted by water availability, being reduced by ~35% in WS compared to WW treatment (Figure 6D). Root dry mass (RDM) had not been modified by any investigated factor, being maintained stable, but suffering great variations from plant to plant, judging by high values of SE (Figure 6E). Total dry mass (TDM) was reduced under the water shortage conditions in both genders of ‘THB’, but only in H of ‘Candy’ (Figure 6F), as observed in LDM (Figure 6A), since leaf mass had the highest participation in the formation of TDM. The SSD in TDM was expressed in ‘Candy’, but only in the WS treatment, where hermaphrodites were more sensitive to drought than females.

Leaf mass ratio (LMR, fraction of dry mass retained in the leaves in relation to the TDM) was reduced in WS when compared to WW treatment, but only in hermaphrodites of both varieties (Figure 6G). This means that hermaphrodites were more sensitive to drought than females when LMR was the trait of analysis. Root to shoot ratio (RShR) was lower in WW than in WS treatment (Figure 6H), which was due to similar RDM accumulated in the experimental period under the two water conditions (Figure 6E), together with a higher LDM (Figure 6C) and SDM (Figure 6D) in WW than in WS treatment.

As a synthesis of our results, the short-term water stress provoked differences in the responses of 18 out of the 19 investigated traits, which were observed during one or various days of observations, when WS treatment was compared to WW conditions (Appendix A). The only exception was the RDM, which was observed only at the end of the fourth day after the WS interruption. Additionally, among 52 executed ANOVAs, water availability did not have an impact in only seven cases (as a simple effect or in interactions). As this study had SSDs of young papaya plants as a focus, this phenomenon was expressed together with significant water availability variations. There was only one exception: SRPI gender segregation on the second day after the WS interruption, when the water availability did not vary. In WS treatment, the SSD was expressed in 14 of 18 traits investigated, while in WW treatment, in only 7 of 18 traits.

## 3. Discussion

Our work is the first to report the spectral leaf indices, as well as stomatal conductance and morphology/biomass responses, as possible tools in early gender segregation of papaya hermaphrodites from females. It underlined that a short period of suspension promoted such segregation in later estimations. Some parameters segregated the two genders even under the well-watered conditions, which are desirable conditions for seedling production. Additionally, this work showed the importance of under-species analysis, as the two investigated varieties could diverge and show opposite gender segregation expressed in the values of some observed traits.

### 3.1. Early Expression of SSD in Stomatal Conductance Related to Water Shortage and Variety of Papaya Seedlings

Stomatal regulation controls most leaf gas exchanges, strongly related to carbon acquisition [38]. Through stomatal conductance (*g*_s_), it is possible to express a degree of stomatal closure, which is intrinsically related to the loss of soil hydraulic conductivity. A decrease in *g*_s_ under WS was observed only in ‘THB’ but not in ‘Candy’ (Figure 1). Additionally, *g*_s_ values were reduced in both water availability conditions and varieties, from the first to the fourth day of evaluation, even under WW conditions (Figure 1). This gradual reduction in *g*_s_ can be explained by the high atmospheric water demand expressed through high VPD_air_ and, additionally, by reduced PPFD (Figure 7). A high VPD_air_ can reduce *g*_s_, which is associated with peristomal evaporation [39,40]. Interestingly, *g*_s_ reduction was less expressed in WS than in WW treatment (Figure 1). The *g*_s_ reduction was more expressive in the ‘THB’ than the ‘Candy’ variety under both water availabilities, and the *g*_s_ of ‘THB’ was additionally reduced by WS conditions, indicating this variety as sensitive to short WS conditions in the early seedling stage.

The *g*_s_ differs between the genders of different species, as in the case of *Juniperus thurifera* [13], *Ginkgo biloba* [23], *Salix glauca* [18], or *Ilex paraguariensis* [21]. In these species, the female plants present higher *g*_s_ in relation to the male plants, which is related to the general female strategy of superior *A*_net_ than in males, due to their additional reproductive efforts [6,22], especially in adult plants [20,21]. In some studies, the SSD in *g*_s_ was only observed under stressful conditions [18] or under some phenological stages, such as the early vegetative and early reproductive stages [21]. Currently, contradictory results about SSD in *g*_s_ are obtained in young papaya, where the *g*_s_ does not differ between the genders [41] or is found higher in female than in hermaphrodite adult plants [37]. In the actual experiment, young females showed a lower *g*_s_ than hermaphrodites, but only on the first assessed day (42 DAS) and only in the ‘THB’ variety sensitive to WS (Figure 1). Such quick SSD responses after the drought interruption can be related to the previously exposed fact that *g*_s_ is a primary driver of stomatal closure and an indicator of soil water drying [42].

### 3.2. SSD in Pigment Content Indicators Related to Water Shortage and Variety of Papaya Seedlings

The absolute quantities of the pigments and their ratios are physiological traits that vary according to species, growth/development of the plant, and the environmental conditions [43], such as water availability in the soil [44]. Since the foliar pigment contents are affected by a variety of stress factors, a leaf spectral reflectance analysis can show the degree to which plants are affected by stresses [45,46]. Over the four observed days after water shortage, all pigments associated spectral leaf indices were impacted by water availability (Figure 2 and Figure 3), some starting from the first assessed day (SPAD, CNDVI, CRI1, CRI2, and FRI), and all suffered the modifications on the fourth assessed day, much later than *g*_s_ (Figure 1). The pigment-associated spectral leaf indices were also modified by variety and/or gender (Figure 2 and Figure 3). Let us discuss them separately as Chl and Car or flavonol-related.

Relative Chl concentration (SPAD index), together with spectral properties indicating Chl content (G index and CNDVI), were modified by WS conditions (Figure 2). Leaf pigment degradation occurs in various cultivated plant species subjected to WS [47,48], explaining such a Chl reduction. The Chl-related leaf indices (SPAD, G index, and CNDVI) expressed SSD under the WS conditions on the fourth day after the water-shortage interruption (Figure 2). SPAD segregated F (higher) from H (lower values) seedlings only in ‘Candy’ in WS treatment (Figure 2A), as similarly observed in CNDVI, but in both varieties (Figure 2C), while the G index segregated F from H in both varieties (Figure 2B, Figure 8). Higher values of the G index were found in H than in F seedlings, not converging with SSD expressions in SPAD or CNDVI. Such opposite gender responses of the G index compared to CNDVI or SPAD can be explained by the fact that the G index is sensitive to leaf area [49], CNDVI focuses on the interaction between Chl and N [50], while SPAD measures light transmission through the leaf to estimate the Chl content [51]. Also, those three indices use different wavelength ranges. Females of *Ginkgo biloba* [23] and *Salix paraplesia* [52] show higher Chl content than males, similar SSD relations to our results of SPAD or CNDVI, but herein segregating females from hermaphrodites. Opposite results are also found in some other dioic species, with males of *Juniperus communis* [9] or *Salix purpurea* [53] having a higher SPAD compared to females under the WS. In adult papaya plants, cultivated in the field under the WW conditions, a lower concentration of Chl is found in F than in the H plants [37], as similarly shown here in G index responses under WS (Figure 2B). Our results suggested that the G index, CNDVI, and SPAD (Chl content indicators) can be used for gender segregation of young papaya plants, where the moment of such SSD expression estimation was essential, here four days after the water-shortage interruption.

Carotenoids and flavonoids play important roles in protection against various stresses and in drought resistance of higher plants [54,55]. Flavonoids act as complements to other reactive-oxygen-species-scavenging systems. Despite their protective role, here, the investigated indicators of Car (CRI1 and CRI2) and flavonoid (FRI) contents diminished values from the first to the fourth day after the water-shortage interruption (Figure 3), suggesting their decomposition with time after the short duration of water suspension. Such diminished index values were also observed under WW conditions. Even though the WW papaya seedlings absorbed more water, they also experienced moderate water shortage, bearing in mind the size of the tubes used in the experiment. The reduction in CRI1, CRI2, and FRI values under WW can be related to a little substrate that filled the tubes and, consequently, substrate fast drying, due to the intense water demand in the air in the conditions of this experiment, indicated by elevated VPD_air_ (Figure 7H) and by reduced PPFD (Figure 7E).

The flavonoid-associated index (FRI) was very sensitive to water shortage in papaya seedlings, being much higher in WS than in WW treatment, indicating that secondary scavenging systems were activated [54], but FRI did not show any SSD expression (Figure 3C). Car-associated indices expressed the SSD in both varieties on the fourth day after the water-shortage interruption (Figure 3A,B). The SSD was expressed in those indices even under WW conditions. On the fourth day after the water-shortage interruption, females of ‘Candy’ had higher CRI1 than hermaphrodites, irrespective of water conditions, while ‘THB’ hermaphrodites had higher CRI1 than females, but only in WS treatment (Figure 3A). The CRI2 was even more expressive in gender segregation, which started from the third day after the water-shortage interruption, when F showed higher values than H plants under WW, with a similar situation in ‘Candy’ under WS, while ‘THV’ had the opposite SSD response under WS condition, H being of higher CRI2 than F plants (Figure 3B). As CRI1 and CRI2 are shown to be very sensitive to drought [56] and can be used in weed detection in the field [57], their spectra of utilities can be amplified by a recommendation of gender segregation, at least in young papaya plants under both WW and WS conditions.

### 3.3. SSD in Water Indicators Related to Water Shortage and Variety of Papaya Seedlings

Here, we analyzed two indices related to water status, PRI and WBI. Spectral signals associated with plant water status are more evident at canopy than leaf scale [58] and can be related to different processes. For example, at the canopy scale in maize, PRI can indicate water stress in young plants before the leaf area increases [59,60], while at leaf/plant scale, it is related to the de-epoxidation cycle of xanthophyll and to heat dissipation that increases under WS conditions [61]. The second water index, WBI, provides information about changes in the relative water content and leaf water potential, even in *g*_s_, especially at canopy scale [58,62].

The WBI values observed at plant scale of papaya were reduced under water shortage, at the first and last observed days (Figure 4B). Its values dropped gradually over the evaluation period under both WW and WS conditions, which could be explained by elevated VPD_air_ (Figure 7H) and small soil substrate quantity, as discussed related to *g*_s_, Car or flavonoid content indicators (Figure 1 and Figure 3). On the 4th observed day, two papaya genders strongly differed in WBI, which was always higher in females than hermaphrodites, irrespective to water conditions. On the other hand, PRI did not show the gradual reduction over the observed period but also expressed SSD only under WS conditions on the 4th observed day, also with females being less impacted by drought than hermaphrodites. Two genders of *Honckenya peploides* show significant PRI time variation (as papaya here, Figure 3A), but do not express SSD [63], while SSD in PRI was found in papaya. Time variation in PRI and WBI is also registered in dioecious *Populus angustifolia*, where SSD in PRI is reported earlier in autumn than SSD in WBI [64]. Interestingly, papaya variety ‘THB’ did not show the reduction in WBI or PRI values under the WS when compared to WW, while those two water status indicators were significantly reduced in ‘Candy’ in WS. Such variety responses, together with indices previously related to pigment contents, clearly indicated that the gender segregation and water stress responses in papaya must be analyzed always at variety scale, but do not exclude the possibility that other varieties would show similar SSD expressions.

### 3.4. SSD in Structural Indicators Related to Water-Shortage and Variety of Papaya Seedlings

Here, we analyzed SSD expressions in two spectral structural indices, SIPI and SRPI, and both were impacted by water, variety, and gender modifications (Figure 5). Interestingly, among all indices, SRPI was the earliest one expressing SSD, on the second day after the water-shortage interruption, with higher values in hermaphrodites than in females, irrespective of variety or water condition, and such gender segregation was lost in the subsequent days of observation (Figure 5B). As SRPI is correlated to plant N status on large-area monitoring, it is more suitable for fields [65], but even some structural indices segregate two genders in N concentrations analyzed at leaf/plant scale in *Hyeronima alchorneoides* and *Virola koschnyi*, two neotropical dioecious tree species [64]. SIPI expresses the Car/Chl ratio, which is an indicator of the degree of induced stress changes in plants, related to the fact that the degradation of Chl is faster than that of Car under stress conditions [65]. The SIPI was generally lower in WS than in WW in papaya seedlings with some exceptions (Figure 5A). The subspecies SSD in SIPI was underlined, because this index differed only in ‘Candy’ and not in ‘THB’, being higher in females than hermaphrodites on the fourth day after the water-shortage interruption, irrespective of water availability. Also, SIPI was higher in ‘Candy’ than in ‘THB’ under WS conditions, indicating that Chl and Car degradation was faster in ‘THB’ than in ‘Candy’ under the WS. The Chl-content-associated indices did not indicate such a trend (Figure 2), but the Car-associated indices did (Figure 3). As a conclusion about SSD in the structural indices of young papaya, SIPI can be used for gender segregation, but only in ‘Candy’ under any water condition.

### 3.5. SSD in Morphological Traits and Biomass Accumulation Related to Water Shortage and Variety of Papaya Seedlings

Under water shortage, decreased growth is a visible response, occurring due to the reduction in turgidity, which decreases the cell expansion rate and net CO_2_ assimilation rate controlled by *g*_s_ and stomatal closure [42,66]. Following our hypothesis, such conditions would promote SSD. All morphological and biomass traits related to leaf formation and leaf biomass accumulation, such as LN, LDM, and LMR, expressed the SSD under the WS treatment (Figure 6). On the other hand, stem- and root-related growth and biomass characteristics (SD, SDM, RDM, RShR) did not differ among F and H seedlings. In *Honckenya peploides*, the growth rate of the two genders does not show significant differences, but males show higher LMR and lower *A*_net_ than females, while females show a higher water content and succulence than males [63]. Hermaphrodites of young papaya were shown to be more sensitive to drought than females in all leaf-related traits. Our experiment also indicated that ‘Candy’ expressed SSD in total biomass, due to the great sensitivity of hermaphrodites to drought, but the low sensitivity to drought of females, the latter showing a similar TDM under the WW and WS conditions. In the meantime, both genders of ‘THB’ similarly reduced TDM under the WS. Observing the growth parameters, the recommendation of parameter choice for gender segregation in papaya would be to always relate them to leaf characteristics and to note the differences between the varieties.

## 4. Materials and Methods

### 4.1. Plant Material and Experimental Conditions

The experiment was carried out in a greenhouse on the North Fluminense State University (UENF), Campos dos Goytacazes (21°44′47″ S, 41°18′24″ W and 10 m a.s.l.), RJ, Brazil. The seeds of two papaya varieties, ‘Candy’ and ‘THB’, were sown on 26 January 2022, in 50 cm^3^ plastic test tubes with Basaplant^®^ commercial substrate with the addition of 0.55 g Osmocote^®^ 19-6-10 Mini Prill (3M) to each tube. Seed germination started 15 days after sowing (DAS). Irrigation was performed daily by micro-spraying until a visual soil saturation, five times a day (at 6:00 a.m., 9:00 a.m., midday, 2:00 p.m., and 5:00 p.m.), always until full substrate saturation (about 36.6 ± 3.1% of soil water content, without water limitations, as full field capacity). The experiment scheme is shown in Figure 8.

From a sowing date of March 5, 2022 (38 DAS), about 400 papaya seedlings (200 of each variety) were cultivated under the greenhouse covered with a diffusive film (Electro M30) of 120 µm (https://lojatropicalestufas.com.br/filmes-agricolas/filme-difusor/difusor-120-micras/filme-difusor-120-micras-electro-m30/ (accessed on 19 July 2025)). The microclimate was followed using a data logger (Weather Stations model 2000, Spectrum Technologies, Plainfield, IL, USA). The daily average values of the photosynthetic photon flux density (PPFD, µmol m^−2^ s^−1^), air temperature (°C), relative humidity (RH, %), and air vapor pressure deficit (VPD_air_, kPa) in the greenhouse, in the pre-experimental period (from germination to change in light conditions) attained 1050 µmol m^−2^ s^−1^, 38 °C, 58%, and 4.2 kPa (Figure 7A–D, respectively).

To decrease the evapotranspiration rate, the papaya seedlings were transferred to a bench located inside the greenhouse under one additional net shading that additionally reduced the PPFD by 70% (https://loja.zanatta.com.br/tela-sombrite-70?srsltid=AfmBOoqUQ9jTSESxAmUuUBB2fEf4rmF3KIHMB1iI1eHKOrpwwG8aUhur (accessed on 10 June 2025)). This transfer occurred on 5 March 2022 (38 DAS), permitting stable average daily values of 290 µmol m^−2^ s^−1^ (Figure 7E). In the second light environment, the conditions were better for papaya seedling growth, with a mean daily temperature of 29.5 °C (Figure 7F), and the daily mean values of RH and VPD_air_ were 58% and 2.38 kPa (Figure 7G and Figure 7H, respectively). In this second light environment, the water shortage (or water stress, WS) was applied, starting on 39 DAS, at 5:00 p.m. (indicated with red flash on Figure 7E–H). Spraying restarted on DAS 41 at 2:00 p.m. (blue flashes on Figure 7E–H). Such short-term water suspension had only 45 h of duration, which was the breaking point because visual signs of leaf wilting were observed. Even though the water stress had only 45 h of duration, the soil water content was drastically diminished. The soil water content in three tubes of each variety was determined by carefully separating substrate from roots above Petri dishes after 24 h (being 21.84 ± 9.00% in ‘Candy’ and 21.60 ± 4.55% in ‘THB’) and after 45 h (being 10.09 ± 3.55% in ‘Candy’ and 6.36 ± 2.42% in ‘THB’).

### 4.2. Determinations of Plant Gender

For gender determination, circles of 3 mm in diameter were collected from the leaf blades of the most expanded leaf from each seedling on 29 DAS. The determination was performed by extracting genomic DNA and using PCR amplification. qPCR used in analysis determines primers specific to the Y chromosome in papaya (https://www.linkedin.com/in/mf-papaya-31582321b/?originalSubdomain=br (accessed on 10 June 2025), MF papaya, Campos dos Goytacazes, RJ, Brazil).

After the determination of hermaphrodite (H), female (F), and male plants, for the experiment of ecophysiological gender segregation, 40 H and 40 F seedlings of each of the two varieties were used (totaling 160 seedlings), which were subsequently exposed to WW and WS conditions (Figure 8). In this manner, 20 seedlings of each gender of each variety represented a sampling unit under each of the two water conditions.

### 4.3. Evaluations of Ecophysiological Traits

The *g*_s_, green intensity index (SPAD index), and leaf spectral optical properties were evaluated daily, from 9 to 12 March 2022, corresponding to a period of 42 to 45 DAS (first to fourth day after the WS interruption). All ecophysiological evaluations were performed on the third leaf counted from the seedling basis from 8:00 to 11:00 a.m. The *g*_s_ was determined using a leaf porometer SC–1 and SPAD index, using a handheld device SPAD–502 (Conica Minolta, Japan), while nine optical leaf properties (Table 1) were evaluated by using a CI–710 portable mini leaf spectrometer (CID–BioScience Inc., Camas, WA, USA). The spectrometer was configured to perform reflectance readings, adopting a signal integration time set of 300 ms, integration time factor at the highest value, scan’s average ‘signal: noise’ ratio of 2, and a boxcar width value of 10 [67,68]. Here, we used one classification based on biological interpretations that helped in index presentations (Table 1).

### 4.4. Evaluations of Plant Morphology and Dry Mass Accumulation

At the end of the experiment, the leaf number (NL) was counted, and the stem diameter (SD) was measured with digital vernier calipers (JOMARCA, Guarulhos, SP, Brazil) at each seedling. After morphological evaluations, the leaves, stem, and roots were separated and placed in the paper bags for drying using forced-air circulation in an oven model 320/5 (Eletrolab, São Paulo, SP, Brazil) at 70 °C for 72 h. The roots were previously washed in running water to remove the substrate. After the drying period in the oven, the dry mass of leaves (LDM), stems (SDM), roots (RDM), and the total plant dry mass (TDM) were determined using a Shimadzu AY220 precision balance (Shimadzu Corporation, Kyoto, Japan).

To assess whether the investment was greater for roots or shoots, the root-to-shoot ratio (RShR) was calculated as follows: RShR = RDM/(LDM + SDM). The leaf mass ratio (LMR) was evaluated, corresponding to the fraction of dry mass retained in the leaves in relation to the TDM, and expressing the fraction of non-exported DM from leaves to other organs (LMR = LDM/TDM).

### 4.5. Statistical Analysis and Experimental Design

The completely random factorial experimental design was performed, including three factors: water availability (WW and WS), varieties (‘Candy’ and ‘THB’), and genders (F and H). The responses of ecophysiological parameters were analyzed separately for each of the four days (42 to 45 DAS) that followed the WS interruption, while the morphological and biomass parameters were analyzed only once at the end of the experiment (45 DAS).

ANOVAs were performed using the ‘R’ software, R.4.2.1 [72]. Three-way ANOVAs (2 × 2 × 2) were processed after the use of mixed linear modeling (lme function and maximum likelihood from ‘nlme’ package), considering water availability, varieties, and genders as fixed factor effects, while plant number (repetition) was a random effect (n = 20). If no significant interaction (starting from the most complex one, where three factors are interacting) was found, the model reduction was applied (and fitted by using the lme function, considering again all factors as fixed or random, as mentioned before). Bartlett's homogeneity and Shapiro's normality tests were performed for each variable in each season. For comparing the average values estimated by ANOVA(s), we used the Tukey HSD and ‘lsmeans’ and ‘multcompView’ packages. A significance level of 0.05 was used for all analyses. The estimated means, standard errors (SE), and *p*-values are shown in charts. If the three-factor interaction was shown to be significant, only the *p*-value of that interaction is shown (as it encompasses all interactions), leaving the figures as unpolluted by information excess as possible.

## 5. Conclusions

We responded positively to both hypotheses, showing that seedling traits, such as *g*_s_, SPAD index, and leaf reflectance indices, can be used in papaya gender segregation, and that SSD was most expressed under the water stress (WS) treatment. Under the WS treatment, the SSD was expressed in 14 of 18 traits investigated: *g*_s_ (H > F in ‘THB’), SPAD (F > H in ‘Candy’), G index (H > F), CNDVI (F > H), CRI1 and CRI2 (F > H in ‘Candy’, and H > F in ‘THB’), PRI (F > H), WBI (F > H), SIPI (F > H in ‘Candy’), SRPI (H > F), LN (F > H), LDM (F > H), TDM (F > H in ‘Candy’), and LMR (F > H). The SSD occurred even under WW conditions in 7 of 18 traits investigated: *g*_s_ (F > H in ‘Candy’), CNDVI (H > F), CRI1 (F > H in ‘Candy’), CRI2 (F > H), WBI (F > H), SIPI (F > H in ‘Candy’), and SRPI (H > F). As tools for SSD expression, *g*_s_ and SRPI must be measured early, on the first or second day after the WS interruption, while the other parameters can be measured on the fourth day after the WS interruption. To avoid different SSD expressions in varieties, the choice of the clearest responses under WS would be the G index, CNDVI, PRI, WBI, SRPI, LN, LDM, and LMR. If the WW conditions are maintained for papaya seedling production, after the light-condition improvement, the recommendation for SSD expression would be the analysis of CNDVI, CRI2, WBI, and SRPI. The non-destructive optical leaf indices segregated papaya hermaphrodites from females under both water conditions and eventually could be adjusted for wide-scale platform evaluations with planned space arrangements of seedlings and sensor sets.

## Figures and Tables

**Figure 1 plants-14-02445-f001:**
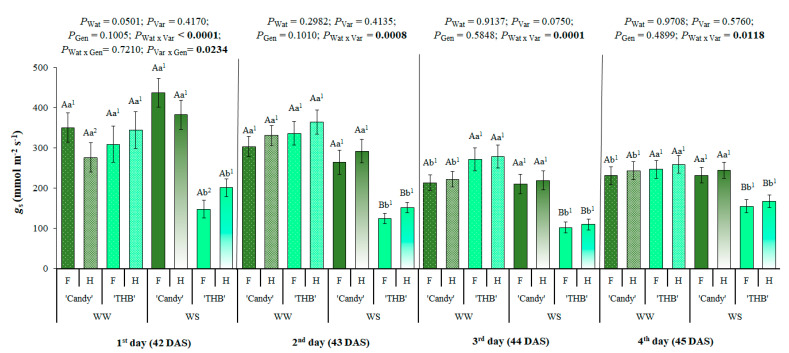
Stomatal conductance (*g*_s_) observed in the period from the first to fourth day after the water-shortage interruption, corresponding to 42 to 45 days after sowing (DAS). Seedlings of two varieties (‘Candy’ and ‘THB’) of *Carica papaya*, including two genders (female: F and hermaphrodite: H), were exposed to well-watered (WW) and water-shortage (WS) conditions from 39 to 41 DAS, and, afterward, all seedlings were well-irrigated. Estimated means ± SE and *p*-values (bold when significant) are shown (n = 20). Uppercase letters compare water treatments (Wat) for each variety and gender, for each DAS; lowercase letters compare variety (Var) responses for each water treatment and gender, for each DAS; superscripted numbers compare gender (Gen) responses in each water treatment and variety, for each DAS.

**Figure 2 plants-14-02445-f002:**
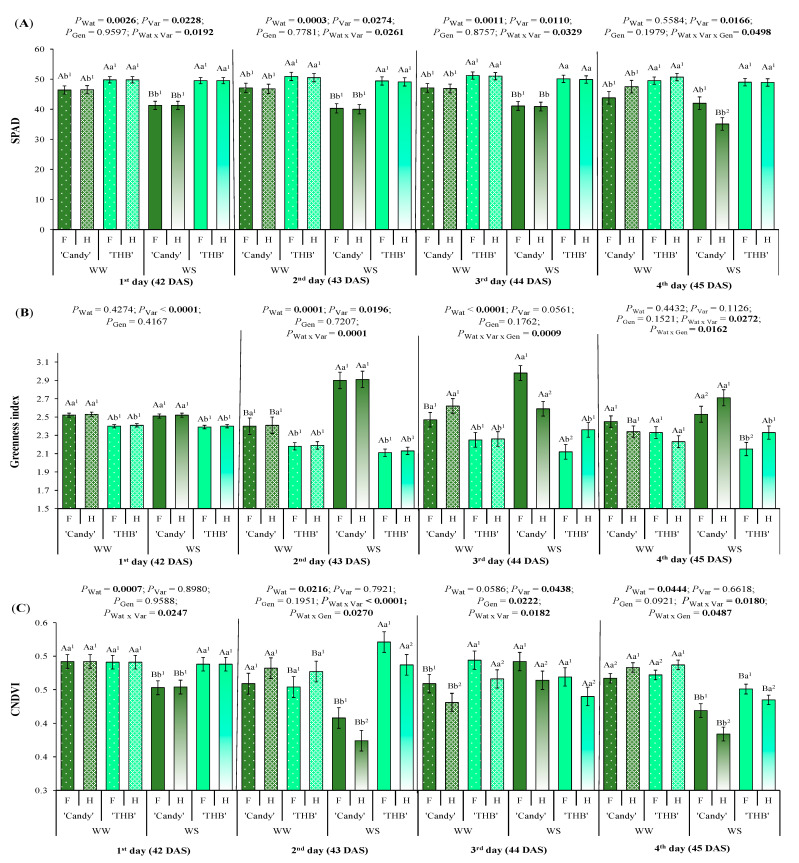
SPAD and spectral optical indices associated with Chl content observed in the period of the first to fourth day after the water-shortage interruption, corresponding to 42 to 45 days after sowing (DAS): (**A**) SPAD, (**B**) greenness index, and (**C**) combination of normalized difference vegetation index (CNDVI). Seedlings of two varieties (‘Candy’ and ‘THB’) of *Carica papaya*, including two genders (female: F and hermaphrodite: H), were exposed to well-watered (WW) and water-shortage (WS) conditions from 39 to 41 DAS, and, afterward, all seedlings were well-irrigated. Estimated means ± SE and *p*-values (bold when significant) are shown (n = 20). Uppercase letters compare water treatments (Wat) for each variety and gender, for each DAS; lowercase letters compare variety (Var) responses for each water treatment and gender, for each DAS; superscripted numbers compare gender (Gen) responses in each water treatment and variety, for each DAS.

**Figure 3 plants-14-02445-f003:**
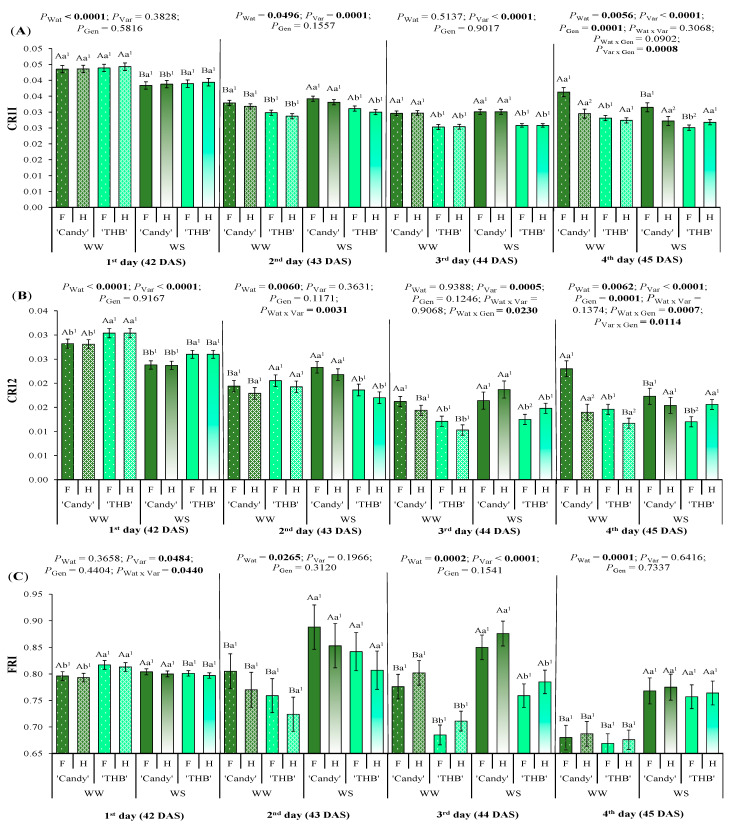
Leaf reflectance indices associated with the pigment contents observed in the period of the first to fourth day after the water-shortage interruption, corresponding to 42 to 45 day after sowing (DAS): (**A**) carotenoid reflectance index 1 (CRI1), (**B**) carotenoid reflectance index 2 (CRI2), and (**C**) flavonoid reflectance index (FRI). Seedlings of two varieties (‘Candy’ and ‘THB’) of *Carica papaya*, including two genders (female: F and hermaphrodite: H), were exposed to well-watered (WW) and water-shortage (WS) conditions from 39 to 41 DAS, and, afterward, all seedlings were well-irrigated. Estimated means ± SE and *p*-values (bold when significant) are shown (n = 20). Uppercase letters compare water treatments (Wat) for each variety and gender, for each DAS; lowercase letters compare variety (Var) responses for each water treatment and gender, for each DAS; superscripted numbers compare gender (Gen) responses in each water treatment and variety, for each DAS.

**Figure 4 plants-14-02445-f004:**
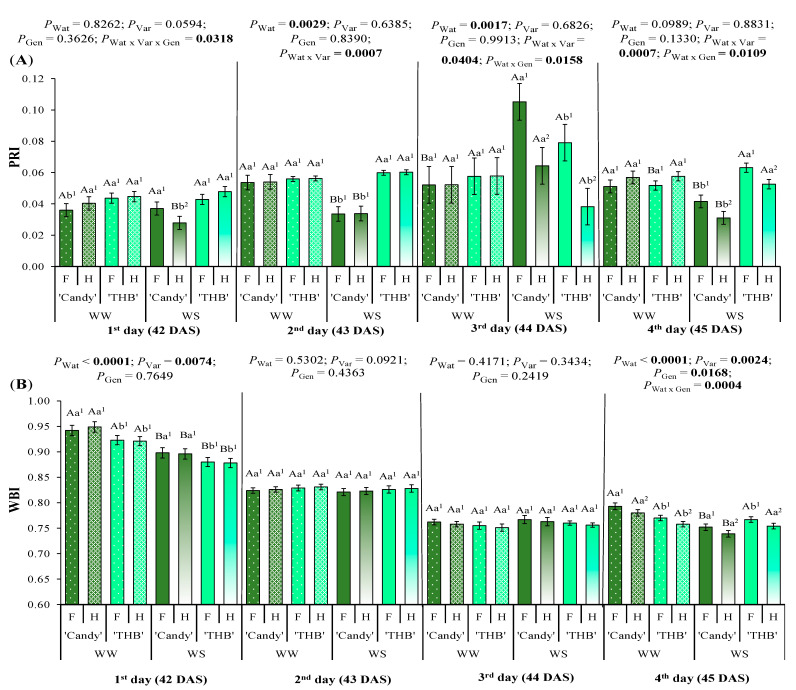
Leaf reflectance indices associated with the water status observed in the period from the first to fourth day after the water-shortage interruption, corresponding to 42 to 45 days after sowing (DAS): (**A**) photochemical reflectance index (PRI) and (**B**) water band index (WBI). Seedlings of two varieties (‘Candy’ and ‘THB’) of *Carica papaya*, including two genders (female: F and hermaphrodite: H), were exposed to well-watered (WW) and water-shortage (WS) conditions from 39 to 41 DAS, and, afterward, all seedlings were well-irrigated. Estimated means ± SE and *p*-values (bold when significant) are shown (n = 20). Uppercase letters compare water treatments (Wat) for each variety and gender, for each DAS; lowercase letters compare variety (Var) responses for each water treatment and gender, for each DAS; superscripted numbers compare gender (Gen) responses in each water treatment and variety, for each DAS.

**Figure 5 plants-14-02445-f005:**
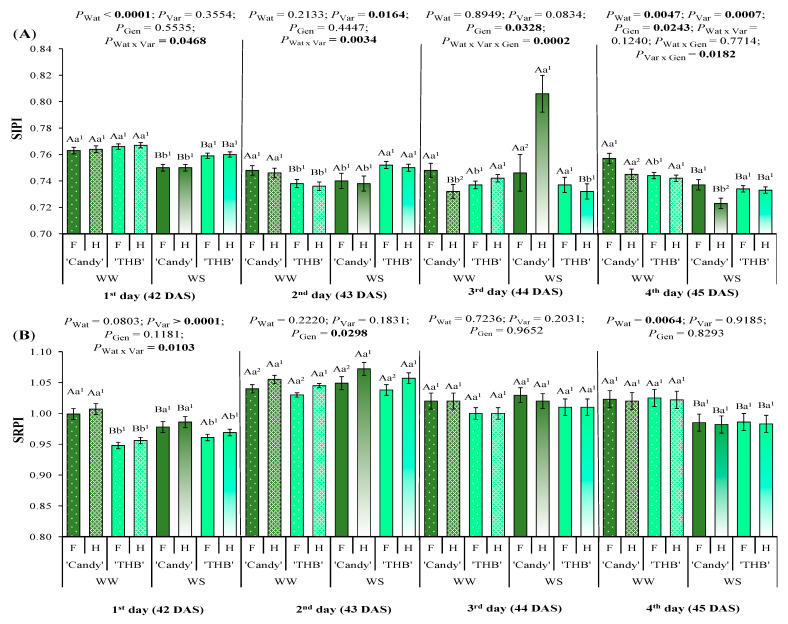
Leaf reflectance indices associated with the structure observed in the period from the first to fourth day after the water-shortage interruption, corresponding to 42 to 45 days after sowing (DAS): (**A**) structure-insensitive pigment index (SIRI) and (**B**) simple ratio pigment index (SRPI). Seedlings of two varieties (‘Candy’ and ‘THB’) of *Carica papaya*, including two genders (female: F and hermaphrodite: H), were exposed to well-watered (WW) and water-shortage (WS) conditions from 39 to 41 DAS, and, afterward, all seedlings were well-irrigated. Estimated means ± SE and *p*-values (bold when significant) are shown (n = 20). Uppercase letters compare water treatments (Wat) for each variety and gender, for each DAS; lowercase letters compare variety (Var) responses for each water treatment and gender, for each DAS; superscripted numbers compare gender (Gen) responses in each water treatment and variety, for each DAS.

**Figure 6 plants-14-02445-f006:**
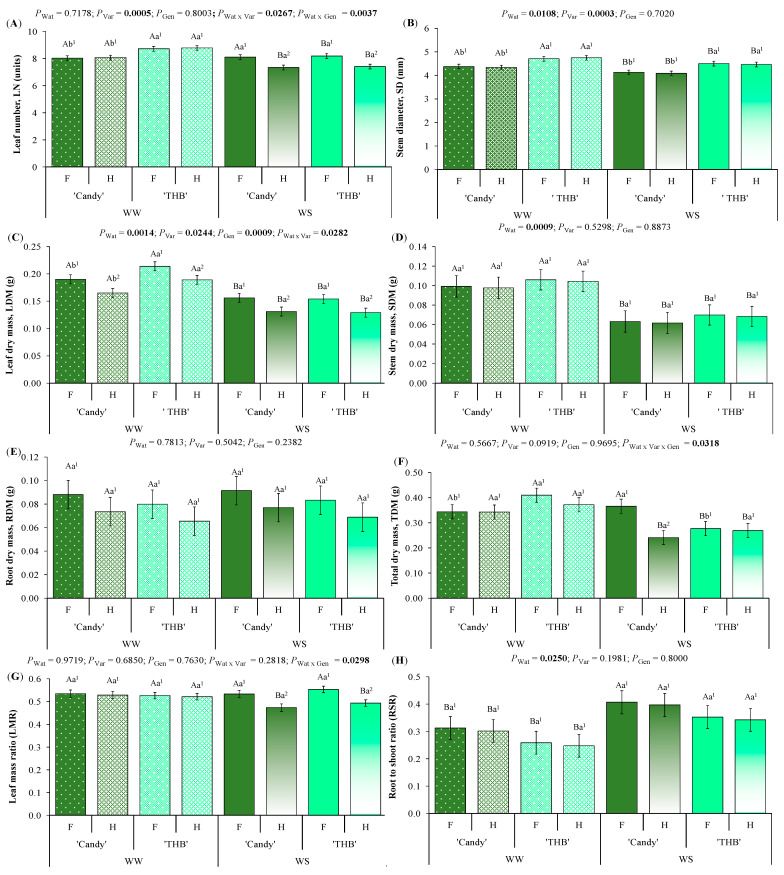
Morphological traits and biomass accumulation in seedlings observed at the end of experiment: (**A**) leaf number, (**B**) basal stem diameter, (**C**) leaf dry mass, (**D**) stem dry mass, (**E**) root dry mass, (**F**) total dry mass, (**G**) leaf mass ratio (fraction of dry mass retained in the leaves in relation to the total dry mass), and (**H**) root-to-shoot ratio. Seedlings of two varieties (‘Candy’ and ‘THB’) of *Carica papaya*, including two genders (female: F and hermaphrodite: H), were exposed to well-watered (WW) and water-shortage (WS) conditions from 39 to 41 days after sowing, and, afterward, all seedlings were well-irrigated. Estimated means ± SE and *p*-values (bold when significant) are shown (n = 20). Uppercase letters compare water treatments (Wat) for each variety and gender; lowercase letters compare variety (Var) responses for each water treatment and gender; superscripted numbers compare gender (Gen) responses in each water treatment and variety.

**Figure 7 plants-14-02445-f007:**
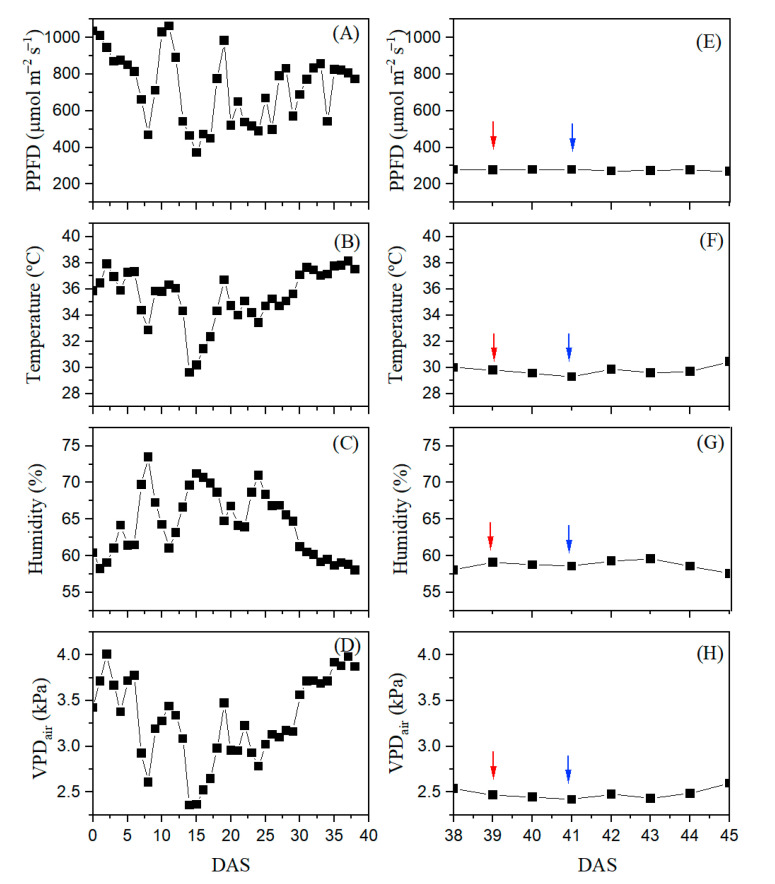
Mean daily photosynthetic photon flux density (PPFD, µmol m^−2^ s^−1^), air temperature (° C), relative humidity (RH, %), and air vapor pressure deficit (VPD_air_, kPa) inside the greenhouse during the pre-experimental period (**A**, **B**, **C**, and **D**, respectively), followed by a short experimental period, in greenhouse with additional net shading (**E**, **F**, **G**, and **H**, respectively). The red and blue flashes indicate the beginning and the end of the period of water suspension, which induced water stress in papaya seedlings.

**Figure 8 plants-14-02445-f008:**
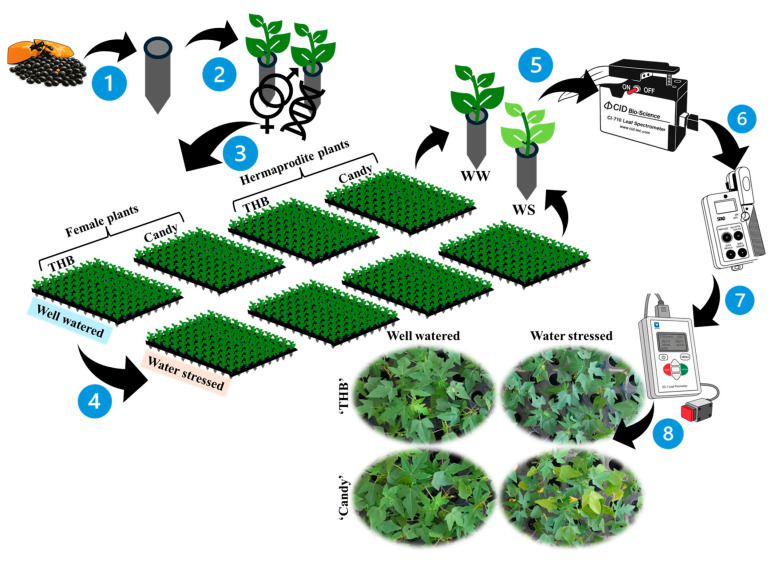
The experimental procedure scheme: (1) Papaya seeds of two varieties; ‘Candy’ and ‘THB’ were seeded (2) in tubes on day after sowing (DAS) = 0. (3) The determination of plant genders was conducted on DAS 29, segregating female from hermaphrodite plants. Steps 1–3 were executed in the greenhouse. (4) The seedlings were transferred under one net (additionally reducing the PPFD by 70%) in the greenhouse on 38 DAS, with the last water spraying effectuated at 5:00 p.m. on 39 DAS in water-shortage (water stress, WS) treatment. The water supply was suspended for 45 h in WS treatment, to 2:00 p.m. on 41 DAS. (5–6) Evaluations of leaf reflectance indices, SPAD index, and (7) stomatal conductance were performed from 42 to 45 DAS, followed by (8) evaluations of morphology and biomass.

**Table 1 plants-14-02445-t001:** The evaluated leaf spectral indices classified by their biological interpretations, with formulas used for calculation (R is reflectance at associated λ).

Index	Full Term	Biological Association	Equation	References
Pigment indices				
G index	Greenness index	Chl content	R_554_/R_677_	[49]
CNDVI	Combination of normalized difference vegetation index	Chl content	(R_750_ − R_705_)/(R_750_ + R_705_)	[50]
CRI1	Carotenoid reflectance index 1	Chl and Car content	(1/R_510_) − (1/R_550_)	[46,69]
CRI2	Carotenoid reflectance index 2	Total Car content	(1/R_510_) − (1/R_700_)	[46,69]
FRI	Flavonoid reflectance index	Flavonol content and screening of excessive visible and UV-A radiation	[(1/R_410_) − (1/R_460_)] − R_800_	[43]
Water indices				
PRI	Photochemical reflectance index	Water status and water stress	(R_531_ − R_570_)/(R_531_ + R_570_)	[70]
WBI	Water band index	Water status and relative water content	R_900_/R_970_	[58]
Structural indices				
SIPI	Structure intensive pigment index	Pest damages through ratio Car/Chl	(R_800_ − R_445_)/(R_800_ + R_680_)	[45]
SRPI	Simple ratio pigment index	Large area monitoring of plants’ N status and ratio Car/Chl	R_430_/R_680_	[45,71]

## Data Availability

The raw data supporting the conclusions of this article will be made available by the authors on request.

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
