# Peer review of "Water Stress Promotes Secondary Sexual Dimorphism in Ecophysiological Traits of Papaya Seedlings"

_plants, 2025, doi:10.3390/plants14152445_

Round 1

Reviewer 1 Report

Comments and Suggestions for Authors

The proposal presented in this research is very interesting from both a basic research perspective and from the practical application of its results. However, it suffers from serious methodological problems that prevent us from understanding the actual responses of seedlings to the sources of variation tested.

Since the availability of contrasting water for seedlings is the factor that would affect their functional and morphological performance, it is important to numerically indicate the moisture content of the substrate under which the functional response of the seedlings subjected to the contrasting water availability treatments was evaluated. This is especially true given that, due to the shape of the containers, the substrate moisture content was a gradient within the container, and the roots could have had sufficient water availability in the short time of water-shortage treatment (3 days).

It is necessary to numerically determine (in water potential or at least as a percentage of water) how water availability for the seedlings varied. Was this water availability uniform in the substrate across the containers and for all experimental units?

The results refer to days after sowing; I believe it would be clearer to refer to them as days after the water-shortage interruption.

The quantification of carotenoid, flavonoid, and chlorophyll content is not described in the materials and methods section.

The results refer to temporal changes of 3 days (42 to 43 to 44 to 45 days after sowing). The authors consider that the moisture availability of the substrate confined in the containers would change sufficiently to be reflected in the functional and morphological responses of the seedlings? Was time included as a source of variation in the ANOVAs? In that case, the correct analysis would be repeated-measures ANOVAs.

Presenting the ANOVA tables would allow for a clearer evaluation of the analysis results.

The results (graphs and probability values) show significant differences inherent almost exclusively to the varieties. In very few cases, there is an effect of the other sources of variation. I believe this is due to the fact that there must have been no changes in moisture availability and also because three days of water-shortage treatment is insufficient time for the functional and, above all, morphological changes in the seedlings to manifest. Unfortunately, I think this last point would be the most difficult to resolve for a subsequent submission. I made some specific observations in the PDF manuscript.

Author Response

The proposal presented in this research is very interesting from both a basic research perspective and from the practical application of its results. However, it suffers from serious methodological problems that prevent us from understanding the actual responses of seedlings to the sources of variation tested.

Authors: Thank you for you revision. We tried to respond to you in the manuscript and here, taking care, point by point. Modifications related to your comments made in the manuscript are marked in red.

Since the availability of contrasting water for seedlings is the factor that would affect their functional and morphological performance, it is important to numerically indicate the moisture content of the substrate under which the functional response of the seedlings subjected to the contrasting water availability treatments was evaluated. This is especially true given that, due to the shape of the containers, the substrate moisture content was a gradient within the container, and the roots could have had sufficient water availability in the short time of water-shortage treatment (3 days).

Authors: We agree with you. The numerical determination of the percentage of water in the soil was added in lines 547-552 in M&M:

Such short-term water suspension had only 45 hours of duration, which was the breaking point because visual signs of leaf wilting were observed. Even though the water-stress had only 45 hours of duration, the soil water content was drastically diminished. Soil water content in three tubes of each variety was determined carefully separating substrate from roots above Petri dishes, after 24 hours (being 21.84 ± 9.00 % in ‘Candy’ and 21.60 ± 4.55 % in ‘THB’) and after 45 hours (being 10.09 ± 3.55 % in ‘Candy’ and 6.36 ± 2.42 % in ‘THB’).

 It is necessary to numerically determine (in water potential or at least as a percentage of water) how water availability for the seedlings varied. Was this water availability uniform in the substrate across the containers and for all experimental units?

Authors: We informed variations in lines 547-552 in M&M, as mentioned above.

The results refer to days after sowing; I believe it would be clearer to refer to them as days after the water-shortage interruption.

Authors: Thank you. All figures were corrected and with that, text in the whole manuscript.

The quantification of carotenoid, flavonoid, and chlorophyll content is not described in the materials and methods section.

Authors: We did not work with chemical identification of carotenoid, flavonoid, and chlorophyll contents, but with light reflectance indices, as described in M&M and in the whole manuscript.

The results refer to temporal changes of 3 days (42 to 43 to 44 to 45 days after sowing).

Authors: We followed the changes during four days after the water stress interruption. The water shortage happened from 39th to 41st DAS, as shown in M&M.

The authors consider that the moisture availability of the substrate confined in the containers would change sufficiently to be reflected in the functional and morphological responses of the seedlings?

Authors: This work was focused on secondary sexual dimorphism. We promoted a short water stress of 45 hours and after that, the seedling morpho-physiology was followed for four days, as described in M&M and repeated in the whole manuscript. Both varieties responded strongly to water-shortage, as can be viewed in photo (included in Figure 7), at the end of experiment, and from Figures 1-6. Very, very strong responses to WS were observed in morphology (see Figure 6). Because of the low quantity of substrate and tube’s size, short water stress was sufficient to promote all reactions, nearly always significant when water availability was analyzed (single effect or interactions, see Lines 334-343).

Authors: We added text that can help reader to understand the resumed responses to WS (Lines 334-343):

As a synthesis of our results, the short-term water stress provoked the differences in responses of 18 from 19 investigated traits, which were observed during one or during various days of observations, when WS treatment was compared to WW conditions (Supplementary Table 1). The only exception was the RDM, trait observed only at the end of the 4th day after WS interruption. Additionally, among 52 executed ANOVAs, water availability did not have impact in only seven cases (as simple effect or in interactions). As this study had SSD of young papaya plants as a focus, this phenomena was expressed together with significant water availability variations. There was the only one exception, SRPI gender segregation in the 2nd day after WS interruption when water availability did not vary. In WS treatment, the SSD was expressed in 14 of 18 traits investigated, while in WW treatment only in 7 of 18 traits.

Was time included as a source of variation in the ANOVAs? In that case, the correct analysis would be repeated-measures ANOVAs.

The time was not observed as a source of variations. We could easily show only the 4th day after the stress interruption. We judged that three-factor ANOVA will be super complicate to interpret, and that the fourth factor will only complicate the understanding and interpretation, which was not simple, judging by misunderstanding along with the comments and recommendations of reviewers. See the complexity in The Supplementary Table 1.

Presenting the ANOVA tables would allow for a clearer evaluation of the analysis results.

Authors: Thank you. ANOVA Table was added in Supplementary material and occupied 6 pages. It helps to resume the data.

The results (graphs and probability values) show significant differences inherent almost exclusively to the varieties.

Authors: Varieties responded differently when secondary sexual dimorphism was searched, while the differences due to water shortage were validated  morphologically and physiologically. Short water stress was sufficient to promote nearly always significant responses when water availability was analyzed (single effect or interactions, see Figures 1 to 6). Please, see again Lines 334-343.

In very few cases, there is an effect of the other sources of variation. I believe this is due to the fact that there must have been no changes in moisture availability and also because three days of water-shortage treatment is insufficient time for the functional and, above all, morphological changes in the seedlings to manifest.

Authors: We promoted water stress of 45 hours and after that, the seedling were followed for four days, as described in M&M and the whole manuscript. WW and WS treatments were very different, as can be viewed in photo (included in Figure 7), at the end of experiment, and with attention is possible to read that varieties responded differently to gender segregation. That was our focus. As told above, short water stress was sufficient to promote nearly always significant responses when water availability was analyzed (single effect or interactions, see Figures 1 to 6). Please, see Supplementary Table 1 and Lines 334-343.

Unfortunately, I think this last point would be the most difficult to resolve for a subsequent submission.

Authors: Differences in water responses were highly manifested, especially in water responses. We searched for gender segregation, and they were not always expressed and not in all responses, but responses to water shortage were clear and clean. We provided the paragraph in Lines 334-343 and Supplementary Table 1 that help to interpret data.

I made some specific observations in the PDF manuscript.

Authors: Thank you, we accepted all suggestions with reserve of means:.

The means and SE were estimated (modelled) based on mixed linear modeling (they were not simply calculated as means from values, but in the context of the whole data, as modelling), and for that reason they must be declared as estimated (or modeled).

We added the full field capacity (Lines 514-517).

The number of seedlings is given on lines 528 and 568-572.

Reviewer 2 Report

Comments and Suggestions for Authors

I found this manuscript very interesting. It provides novel results on an old topic: detection of early signs of sexual detection in plants. In addition to the scientific relevance of your results and conlcusions, your findings are of primal importance for cultivators.

Author Response

Comments and Suggestions for Authors

I found this manuscript very interesting. It provides novel results on an old topic: detection of early signs of sexual detection in plants. In addition to the scientific relevance of your results and conlcusions, your findings are of primal importance for cultivators.

Authors: Thank you very much for your positive evaluation.

Reviewer 3 Report

Comments and Suggestions for Authors

1. How to do “By water stress could be promote secondary sexual dimorphism of papaya “?

2. The reproductive organs formed by many factors not only water, so in the temperature, light—-under the better or best conditions in the research?

3. Please to explain” why analyzed the results of 42 to 45 DAS”?

4. The results showed “that the gender segregation and water stress responses in papaya must be analyzed always at variety scale”, so it could be apply on other variety or not?

5. The study investigated many eco physiological factors (gs,SPAD,PPFD—-), how to be adjusted for wild scale platform evaluations?

Author Response

Comments and Suggestions for Authors

Authors: Thank you for your comments. In the manuscript, the modifications due to your observations are marked in blue.

  1. How to do “By water stress could be promote secondary sexual dimorphism of papaya “?

Authors: The whole manuscript is trying to respond to this question. Stressful conditions can promote the gender segregation in young and adult plants. We tested such hypothesis in very young seedlings, with low leaf number (see Figure 7 and incorporated photo). WS was applied for only 45 hours and seedling morpho-physiology was followed.

  1. The reproductive organs formed by many factors not only water, so in the temperature, light—-under the better or best conditions in the research?

Authors: Well, we introduced the theory of secondary sexual dimorphism.  We did not search for good sexual organ formation, but young seedling segregation based on secondary sexual dimorphism (SSD). SSD characteristics are better segregated under stressful conditions (see Lines 61-799). The light/temperature conditions that were offered to young plants were adequate and usual for papaya growth in early stages.

  1. Please to explain” why analyzed the results of 42 to 45 DAS”?

Authors: Thank you for such observation. Such determination was made because the short water stress was applied on 39 and 41 day after sowing (DAS), and measurements were made from 42 to 45 DAS.

We made the modification all over the text to help the reader understand our experimental dynamics, indicating that measurement were made from the 1st to 4th day after the water-shortage interruption, corresponding to 42 to 45 day after sowing (DAS) (starting from Lines 135, 143-144 and all over the manuscript).

  1. The results showed “that the gender segregation and water stress responses in papaya must be analyzed always at variety scale”, so it could be apply on other variety or not?

Authors: Yes, we are convinced that it is applicable to other varieties, or even species, but varietal response must be followed (Lines 349-351)

  1. The study investigated many eco physiological factors (gs,SPAD,PPFD—-), how to be adjusted for wild scale platform evaluations?

Authors: Do you mean wide platform evaluations? We see the possibility to make the analysis of the cleanest parameters (CNDVI, carotenoid reflectance index 2, WBI and SRPI) using the seedling arrangements in clean lines, permitting that the individual plant leaf responses of leaf reflectance, with sensors can be detected and on that way, the plant gender segregation can be made: (mentioned in Lines 32-40 in Abstract):

… and Lines 637-639: The non-destructive leaf optical indices segregated papaya hermaphrodites from females under both water conditions and eventually could be adjusted for wide scale platform evaluations with planed space arrangements of seedlings, and sensor’s set.

Reviewer 4 Report

Comments and Suggestions for Authors

The whole experimentation makes the manuscript difficult to read for the reader.

Very well written manuscript. In the uploaded version highlighted are the phrases needing improvement.

L512, 526 what is ‘one net’? Describe the net(s)

L539 WS degree, please improve with: progression, intensity, etc.

L537-539 There are some problems with the two sentences. There is no 68.8% RH in the graph and watch out for what is minimum (2.5 kPa) and what is mean values (2.3 kPa)!

L301, 327 leaf mass ratio: clarify ratio of what. You have at M+Ms but here it is also needed, I believe.

L398 ‘in a plant kingdom’: what do you mean?

Author Response

Comments and Suggestions for Authors

The whole experimentation makes the manuscript difficult to read for the reader.

Authors: We tried to make the manuscript clearer, with experimentation that was better explained in some details.

Very well written manuscript. In the uploaded version highlighted are the phrases needing improvement.

Authors: Thank you. We made corrections that were suggested by you here and in pdf version and marked them in violet color.

L512, 526 what is ‘one net’? Describe the net(s)

Authors:

…. covered with diffusive film Electro M30 of 120 µm (https://lojatropicalestufas.com.br/filmes-agricolas/filme-difusor/difusor-120-micras/filme-difusor-120-micras-electro-m30/). (Lines 529-531)

…. that additionally reduced the PPFD for 70% (https://loja.zanatta.com.br/tela-sombrite-70?srsltid=AfmBOoqUQ9jTSESxAmUuUBB2fEf4rmF3KIHMB1iI1eHKOrpwwG8aUhur). (Lines 538-540)

L539 WS degree, please improve with: progression, intensity, etc.

Authors: We improved this part of the text also demanded by reviewer #1.

L537-539 There are some problems with the two sentences. There is no 68.8% RH in the graph and watch out for what is minimum (2.5 kPa) and what is mean values (2.3 kPa)!

Authors: That’s true. Thank you. We improved (Lines 541-544):

This transfer occurred on March 05, 2022 (38 DAS), permitting stable average daily values of 290 µmol m-2 s-1 (Figure 8E). In the second light environment, the conditions were better for papaya seedling growth, with a mean daily temperature of 29.5°C (Figure 8F), and the daily mean values of RH and VPDair of 58% and 2.38 kPa (Figures 8G and 8H, respectively).

L301, 327 leaf mass ratio: clarify ratio of what. You have at M+Ms but here it is also needed, I believe.

Authors: It was added:

…fraction of dry mass retained in the leaves in relation to the total dry mass  (Lines 302-303 and 327-328)

L398 ‘in a plant kingdom’: what do you mean?

Authors: It was changed to:

.. among the plant species (Line 49)

. some other dioic species… (Lines 408-409)